# Fighting Gradients with Gradients: Dynamic Defenses against Adversarial Attacks

## Abstract

Adversarial attacks optimize against models to defeat defenses. Existing defenses are static, and stay the same once trained, even while attacks change. We argue that models should fight back, and optimize their defenses against attacks at test time. We propose dynamic defenses, to adapt the model and input during testing, by defensive entropy minimization (dent). Dent alters testing, but not training, for compatibility with existing models and train-time defenses. Dent improves the robustness of adversarially-trained defenses and nominally-trained models against white-box, black-box, and adaptive attacks on CIFAR-10/100 and ImageNet. In particular, dent boosts state-of-the-art defenses by 20+ points absolute against AutoAttack on CIFAR-10 at $\epsilon_\infty = 8/255$.

## 1 Introduction: Attack, Defend, and Then?

Deep networks are vulnerable to adversarial attacks: input perturbations that alter natural data to cause errors or exploit predictions [54]. As deep networks are deployed in real systems, these attacks are real threats [63], and so defenses are needed. The challenge is that every new defense is followed by a new attack, in a loop [56]. The strongest attacks, armed with gradient optimization, update to circumvent defenses that do not. Such iterative attacks form an even tighter loop to ensnare defenses. In a cat and mouse game, the mouse must keep moving to survive.

Current defenses, deterministic or stochastic, stand still: once trained, they are *static* and do not adapt during testing. Adversarial training [18, 30] learns from attacks during training, but cannot learn from test data. Stochastic defenses alter the network [11] or input [20, 7], but their randomness is independent of test data. Static defenses do not adapt, and so they may fail as attacks update.

Our *dynamic* defense fights adversarial updates with defensive updates by adapting during testing (Figure 1). In fact, our defense updates on every input, whether natural or adversarial. Our defense objective is entropy minimization, to maximize model confidence, so we call our method *dent* for defensive entropy. Our updates rely on gradients and batch statistics, inspired by test-time adaptation approaches [53, 43, 28, 29, 58]. In pivoting from training to testing, dent is able to keep changing, so the attacker never hits the same defense twice. Dent has the last move advantage, as its update always follows each attack.

Dent connects adversarial defense and domain adaptation, which share an interest in the sensitivity of deep networks to input shifts. Just as models fail on adversarial attacks, they fail on natural shifts like corruptions. Adversarial data is a particularly hard shift, as evidenced by the need for more parameters and optimization for adversarial training [30], and its negative side effect of reducing

Submitted to 35th Conference on Neural Information Processing Systems (NeurIPS 2021). Do not distribute.

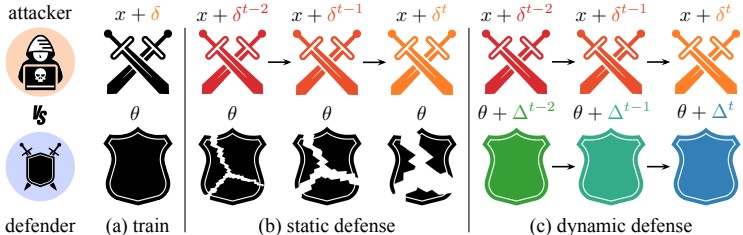

Figure 1: Attacks optimize the input $x + \delta$ against the model $\theta$. Adversarial training optimizes $\theta$ for defense (a), but attacks update during testing while $\theta$ does not (b). Our *dynamic* defense improves robustness by adapting $\theta + \Delta$ during testing (c), so the attack cannot hit the same defense twice.

accuracy on natural data [52, 65]. Faced with these difficulties, we turn to adaptation, and change our focus to testing, rather than training more still.

Experiments evaluate dent against white-box attacks (APGD, FAB), black-box attack (Square), and adaptive attacks that are aware of its updates. Dent boosts state-of-the-art adversarial training defenses on CIFAR-10 by $20+$ points against AutoAttack [9] at $\epsilon_\infty = 8/255$. Ablations inspect the effects of iteration, parameterization, and batch size. Our code is included in the supplement.

**Our contributions**

- We highlight an opportunity for dynamic defense: the last move advantage.

- We propose the first fully test-time dynamic defense: dent adapts both the model and input during testing without needing to alter training.

- Dent augments state-of-the-art adversarial training methods, improving robustness by $30\%$ relative, and tops the AutoAttack leaderboard by $15+$ points.

- We devise two adaptive attacks against dent: denying updates and mixing batches.

## 2   Related Work

**Adversarial Defense** For adaptive adversaries, which change in response to defenses, it is natural to consider dynamic defenses, which adapt in turn. Evans et al. [14] explain dynamic defenses are promising in principle but caution they may not be effective in practice. Their analysis concerns randomized defenses, which do change, but their randomization does not adapt to the input. We argue for dynamic defenses that depend on the input to keep adapting along with the attacks. Goodfellow [17] supports dynamic defenses for similar reasons, but does not develop a specific defense. We demonstrate the first defense to optimize the model and input during testing for improved robustness.

Most defenses for deep learning focus on first-order adversaries [18, 30] that are equipped with gradient optimization but constrained by $\ell_p$-norm bounds. Adversarial training and randomization are the most effective defenses against such attacks, but are nevertheless limited, as they are fixed during testing. Adversarial training [18, 30] trains on attacks, but a different or stronger adversary (by norm or bound) can overcome the trained defense [46, 55]. Randomizing the input [37, 7, 32] or network [11] requires the adversary to optimize in expectation [3], but can still fail with more iterations. Furthermore, these defenses gain adversarial robustness by sacrificing accuracy on natural data. Dent adapts during testing to defend against various attacks without more harm to natural accuracy.

Generative, self-supervised, and certified defenses try to align testing with training but are still static. Generative defenses optimize the input w.r.t. autoregressive [50], GAN [42], or energy [23] models, but the models do not adapt, and may be attacked by approximating their gradients [3]. Self-supervised defenses optimize the input w.r.t. auxiliary tasks [49], but again the models do not adapt. Certified defenses [7, 66] guarantee robustness within their training scope, but are limited to small perturbations by specific types of attacker during testing. Changing data distributions or

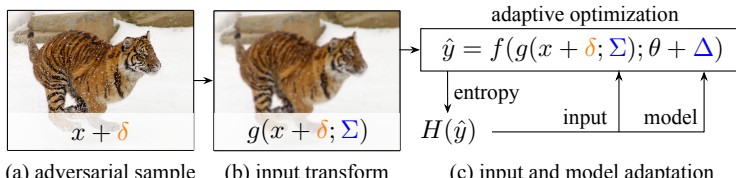

| (a) adversarial sample | (b) input transform | (c) input and model adaptation |

Figure 2: Dent adapts the model and input to minimize the entropy of the prediction $H(\hat{y})$. The model $f$ is adapted by a constrained update $\Delta$ to the parameters $\theta$. The input is adapted by smoothing $g$ with parameters $\Sigma$. Dent updates batch-by-batch during testing.

adversaries requires re-training all of these defenses. Dent adapts during testing, without requiring (re-)training, and is the only method to update the model itself against attack.

**Domain Adaptation** Domain adaptation mitigates input shifts between the source (train) and target (test) to maintain model accuracy [34, 41]. Adversarial attacks are such a shift, and adversarial error is related to natural generalization error [51, 15]. How then can adaptation inform dynamic defense? Train-time adaptation is static, like adversarial training, with the same issues of capacity, optimization, and re-computation when the data/adversary changes. We instead turn to test-time adaptation methods.

Test-time adaptation keeps updating the model as the data changes. Model parameters and statistics can be updated by self-supervision [53], normalization [43], and entropy minimization [58]. These methods improve robustness to natural corruptions [22], but their effect on adversarial perturbations is not known. We base our defense on entropy minimization as it enables optimization during testing without altering model architecture or training (as needed for self-supervision). For defense, we (1) extend the parameterization of adaptation with model and input transformations, (2) optimize for additional iterations, and (3) investigate usage on data that is adversarial, natural, or mixed. We are the first to report test-time model adaptation improves robustness to adversarial perturbations.

**Dynamic Inference** A *dynamic* model conditionally changes inference for each input, while a *static* model unconditionally fixes inference for all inputs. There are various dynamic inference techniques, with equally varied goals, such as expressivity with more parameters or efficiency with less computation. All static models are alike; each dynamic model is dynamic in its own way.

Selection techniques learn to choose a subset of components [1, 57]. Halting techniques learn to continue or end computation [19, 59]. Mixing techniques learn to combine parameters [47, 33, 62]. Implicit techniques learn to iteratively update [6, 4]. While these methods learn to adapt during *training*, our method keeps adapting by directly optimizing during *testing*.

## 3 Dynamic Defense by Test-Time Adaptation

Adversarial attacks optimize against defenses at test time, so defenses should fight back, and counter-optimize against attacks. Defensive entropy minimization (dent) does exactly this for dynamic defense by test-time adaptation.

In contrast to many existing defenses, dent alters testing, but not training. Dent only needs differentiable parameters for gradient optimization and probabilistic predictions for entropy measurement. As such, it applies to both adversarially-trained and nominally-trained models.

### 3.1 Preliminaries on Attacks and Defenses

Let $x \in \mathbb{R}^d$ and $y \in \{1, \ldots, C\}$ be an input sample and its corresponding ground truth. Given a model $f(\cdot; \theta) \colon \mathbb{R}^d \to \mathbb{R}^C$ parameterized by $\theta$, the goal of the adversary is to craft a perturbation $\delta \in \mathbb{R}^d$ such that the perturbed input $\tilde{x} = x + \delta$ causes a prediction error $f(x + \delta; \theta) \neq y$.

A targeted attack aims for a specific prediction of $y'$, while an untargeted attack seeks any incorrect prediction. The perturbation $\delta$ is constrained by a choice of $\ell_p$ norm and threshold $\epsilon$: $\{\delta \in \mathbb{R}^d \mid \|\delta\|_p < \epsilon\}$. We consider the two most popular norms for adversarial attacks: $\ell_\infty$ and $\ell_2$.

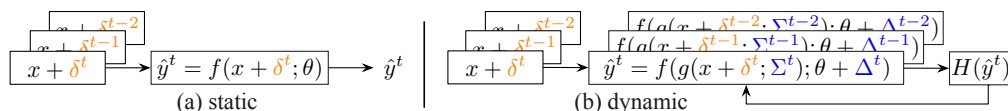

Figure 3: The adversary optimizes its attacks $\delta^{1\cdots t}$ against the model $f$. Static defenses (left) do not adapt, and are vulnerable to persistent, iterative attacks. Our dynamic defenses (right) do adapt, and update their parameters $\Delta, \Sigma$ each time the adversary updates its attack $\delta$.

Adversarial training is a standard defense, formulated by Madry et al. [30] as a saddle point problem,

$$\underset{\theta}{\arg\min}\, \mathbb{E}_{(x,y)} \max_{\delta} L(f(x + \delta; \theta), y), \tag{1}$$

which the model minimizes and the adversary maximizes with respect to the loss $L(\hat{y}, y)$, such as cross-entropy for classification. The adversary iteratively optimizes $\delta$ by projected gradient descent (PGD), a standard algorithm for constrained optimization, for each step $t$ via

$$\delta^t = \Pi_p(\delta^{t-1} + \alpha \cdot \text{sign}(\nabla_{\delta^{t-1}} L(f(x + \delta^{t-1}; \theta), y))), \tag{2}$$

for projection $\Pi_p$ onto the norm ball for $\ell_p < \epsilon$, step size hyperparameter $\alpha$, and random initialization $\delta^0$. The model optimizes $\theta$ against $\delta$ to minimize the loss of its predictions on perturbed inputs. This is accomplished by augmenting the training set with adversarial inputs from PGD attack.

Adversarial training is state-of-the-art, but static. Dynamic defenses offer to augment its robustness.

## 3.2 Defensive Entropy Minimization

Defensive entropy minimization (dent) counters attack updates with defense updates. While adversaries optimize to cross decision boundaries, entropy minimization optimizes to distance predictions from decision boundaries, interfering with attacks. As the adversary optimizes its perturbation $\delta$, dent optimizes its adaptation $\Delta, \Sigma$. Figure 2 shows dent's model ($\Delta$) and input ($\Sigma$) updates.

Dent is dynamic because both $\Delta, \Sigma$ depend on the testing data, whether natural $x$ or adversarial $x + \delta$. On the contrary, static defenses depend only on training data through the model parameters $\theta$. Figure 3 contrasts static and dynamic defenses across the steps of attack optimization.

**Entropy Objective** Test-time optimization requires an unsupervised objective. Following tent [58], we adopt entropy minimization as our adaptation objective. Specifically, our defense objective is to minimize the Shannon entropy [45] $H(\hat{y})$ of the model prediction during testing $\hat{y} = f(x; \theta)$ for the probability $\hat{y}_c$ of class $c$:

$$H(\hat{y}) = - \sum_{c \in 1, \ldots, C} p(\hat{y}_c) \log p(\hat{y}_c) \tag{3}$$

**Adaptation Parameters** Dent adapts the model by $\Delta$ and input by $\Sigma$ (Figure 2). For the model, dent adapts affine scale $\gamma$ and shift $\beta$ parameters by gradient updates and adapts mean $\mu$ and variance $\sigma^2$ statistics by estimation. These are a small portion of the full model parameters $\theta$, in only the batch normalization layers [25]. However, they are effective for conditioning a model on changes in the task [33] or data [43, 58]. For the input, dent updates Gaussian smoothing $g$ by gradient updates of the parameter $\Sigma$, while adjusting the filter size for efficiency [48]. This controls the degree of smoothing dynamically, unlike defense by static smoothing [7].

In standard models the scale $\gamma$ and shift $\beta$ parameters are shared across inputs, and so adaptation updates batch-wise. For further adaptation, dent can update sample-wise, with different affine parameters for each input. In this way, it adapts more than prior test-time adaptation methods with batch-wise parameters [58, 43].

Our model and input parameters are differentiable, so end-to-end optimization coordinates them against attacks as layered defenses. This coordination is inspired by CyCADA [24], for domain adaptation, but dent differs in its purpose and its unified loss. CyCADA also optimizes input and model transformations but does so in parallel with separate losses. Our defensive optimization is joint and shares the same loss.

Table 1: Dent boosts the robustness of adversarial training on CIFAR-10 against AutoAttack. Adversarial training is static, but dent is dynamic, and adapts during testing. Dent adapts batch-wise, while dent+ adapts sample-wise, surpassing the state-of-the-art for static defense at robustbench.github.io.

| ACCURACY(%) | NATURAL | ADVERSARIAL | | |
| --- | --- | --- | --- | --- |
| | | STATIC | DENT | DENT+ |
| $\epsilon_\infty = 8/255$ | | | | |
| CARMON ET AL. [5] | 89.6 | 59.5 | 74.7 | **82.3** |
| SEHWAG ET AL. [44] | 84.4 | 54.4 | 61.2 | **75.2** |
| WONG ET AL. [60] | 83.3 | 43.2 | 52.3 | **71.8** |
| DING ET AL. [12] | 88.0 | 41.4 | 47.6 | **64.4** |
| $\epsilon_2 = 0.5$ | | | | |
| SEHWAG ET AL. [44] | 89.5 | 73.4 | 77.8 | **85.7** |
| RICE ET AL. [38] | 88.7 | 67.7 | 69.7 | **81.3** |
| RONY ET AL. [39] | 89.1 | 66.4 | 73.4 | **85.3** |
| DING ET AL. [12] | 88.0 | 66.1 | 70.3 | **82.8** |

**Update Algorithm** In summary, when the adversary attacks with perturbation $\delta^t$, our dynamic defense reacts with $\Sigma^t, \Delta^t$. The parameters of the model $f$ and smoothing $g$ are updated by $\mathrm{argmin}_{\Sigma,\Delta} H(f(g(x + \delta; \Sigma); \theta + \Delta))$ through test-time optimization. At each step, dent estimates the normalization statistics $\mu, \sigma$ and then updates the parameters $\gamma, \beta, \Sigma$ by the gradient of entropy minimization. Figure 3 contrasts static defenses and dynamic defenses that update like dent.

Dent adapts on batches rather than samples. Batch-wise adaptation stabilizes optimization for entropy minimization. The defense parameters reset between batches.

**Discussion** The purpose of a dynamic defense is to move when the adversary moves. When the adversary submits an attack $x + \delta^t$, the defense counters with $\Delta^t$. In this way, the defense has the last move, and therefore an advantage.

Our dynamic defense changes the model, and therefore its gradients, but differs from gradient obfuscation [3]. Our defense does not rely on (1) shattered gradients, as the update does not cause non-differentiability or numerical instability; (2) stochastic gradients, as the update is deterministic given the input, model, and prior updates; nor (3) exploding/vanishing gradients, as the update improves robustness with even a single step (although more steps are empirically better).

Dent forces the attack to rely on a *stale* gradient, as $\delta^t$ follows $\Delta^{t-1}$, while the model adapts by $\Delta^t$.

# 4 Experiments

We evaluate dent against white-box, black-box, and adaptive attacks with a variety of static defenses and datasets. For attacks, we choose the AutoAttack [9] benchmark, which includes four attack types spanning white-box/gradient and black-box/query attacks. For static defenses, we choose strong and recent adversarial training methods, and we also experiment with nominally trained models. For datasets, we evaluate dent on CIFAR-10/CIFAR-100 [27], as they are popular datasets for adversarial robustness, and ImageNet [40], as it is a large-scale dataset.

We ablate the choice of model/input adaptation, parameterization, and the number of updates.

## 4.1 Setup

**Metrics** We score natural accuracy on the regular test data $x$ and adversarial accuracy on the perturbed test data $x + \delta$. Each is measured as percentage accuracy (higher is better). We report the worst-case adversarial accuracy across attacks.

**Test-time Optimization** We optimize batch-wise $\Delta$ (dent) and sample-wise $\Delta$ (dent+). Dent updates by Adam [26] with learning rate 0.001. Dent+ updates by AdaMod [13] with learning rate 0.006. $\Sigma$

updates use learning rate 0.25. All updates use batch size 128 and no weight decay. Dent+ regularizes updates by information maximization [16, 29]. We tuned update hyperparameters against PGD attacks. Please see the code for exact settings.

**Architecture** For comparison with existing defenses, we keep the architecture and training the same, and simply load the public reference models provided by RobustBench [10]. For analysis and ablation experiments, we define a residual net with 26 layers and a width multiplier of 4 (ResNet-26-4) [21, 64], following prior work on adaptation [53, 58].

## 4.2 Attack Types & Threat Model

We evaluate standard white-box and black-box attacks with adversarially-trained models (Section 4.3) and nominally-trained models (Section 4.4), as well as dent-specific adaptive attacks (Section 4.5).

We primarily evaluate against AutoAttack's ensemble of:

1. APGD-CE [30, 9], an untargeted white-box attack by cross-entropy,

2. APGD-DLR [9], a targeted white-box attack with a shift and scale invariant loss,

3. FAB [8], a targeted white-box attack for minimum-norm perturbation,

4. Square Attack [2], an untargeted black-box attack with square-shaped updates.

These attacks are cumulative, so a defense is only successful if it holds against each type. Following convention, we evaluate $\ell_\infty$ attacks with $\epsilon_\infty = 8/255$ and $\ell_2$ attacks with $\epsilon_2 = 0.5$. This is the standard evaluation adopted by the popular RobustBench benchmark [10].

We devise and experiment with two adaptive attacks against dent and its dynamic updates. The first interferes with adaptation by denying updates: it optimizes offline against $\theta$ without $\Delta, \Sigma$ updates. The second interferes with adaptation by mixing data: it combines adversarial data and natural data in the same batch. Both are specific to dent to complement our general evaluation by AutoAttack.

These attacks fall under the usual white-box threat model. The adversary has full access to the classifier, including its architecture and parameters, and the defense, such as dent's adaptation parameters and statistics. With this access the adversary chooses an attack for each input, but it cannot choose the inputs (the test set is fixed).

We include one additional requirement: dent assumes access to test *batches* rather than individual test *samples*. While independent, sample-wise defense is ideal for simplicity and latency, batch processing is not impractical. For example, cloud deployments of deep learning batch inputs for throughput efficiency, and large-scale systems handle many inputs per unit time [31].

The supplementary material covers more attacks, including AutoAttack Plus and Boundary, to confirm that AutoAttack is a sufficient measure of robustness.

## 4.3 Dynamic Defense of Adversarial Training

We extend static adversarial training defenses with dynamic updates by dent. Compared to nominal training, adversarial training achieves higher adversarial accuracy but lower natural accuracy. The purpose of dent is to improve adversarial accuracy without further harming natural accuracy.

**Dent improves state-of-the-art defenses.** Table 1 shows state-of-the-art adversarial training defenses [5, 44, 39, 38, 60, 12] with and without dynamic defense by dent. Note that dent does not specialize to the choice of norm or bound, unlike adversarial training, but instead adapts to each attack during testing. In each case, dent significantly improves adversarial accuracy and maintains natural accuracy.

Dent updates batch-wise for 30 steps. Dent+ is more robust in fewer steps by sample-wise adaptation. With sample-wise $(\gamma, \beta)$ parameters, dent+ needs only six steps to reach an adversarial accuracy within 90% of the natural accuracy. These experiments only include model adaptation of $\Delta$, without input adaptation of $\Sigma$, as we found it unnecessary when combined with adversarial training.

Table 2: AutoAttack includes four attack types, and dent improves robustness to each on CIFAR-10 against $\ell_\infty$ attacks. We evaluate without dent (-) and with dent (+).

| ACCURACY(%) | APGD-CE | | APGD-DLR | | FAB | | SQUARE | |
|---|---|---|---|---|---|---|---|---|
| | - | + | - | + | - | + | - | + |
| WONG ET AL. [60] | 45.9 | 57.6 | 43.2 | 52.3 | 43.2 | 52.3 | 43.2 | 52.3 |
| DING ET AL. [12] | 50.1 | 60.2 | 41.6 | 48.0 | 41.5 | 47.7 | 41.4 | 47.6 |

Table 3: Ablation of model adaptation ($\Delta$), input adaptation ($\Sigma$), and steps on the accuracy of a nominally-trained model with dent.

| $\Delta$ | $\Sigma$ | STEP | TIME | NATURAL | ADVERSARIAL | |
|---|---|---|---|---|---|---|
| | | | | | $\epsilon_\infty = \frac{1.5}{255}$ | $\epsilon_2 = 0.2$ |
| $\times$ | NONE | 0 | 1.0$\times$ | 95.6 | 8.8 | 9.2 |
| $\sqrt{}$ | NONE | 1 | 3.6$\times$ | 95.6 | 15.0 | 13.5 |
| $\times$ | STAT. | 0 | 1.0$\times$ | 86.2 | 25.8 | 23.6 |
| $\sqrt{}$ | STAT. | 1 | 3.6$\times$ | 86.3 | 27.5 | 24.4 |
| $\sqrt{}$ | STAT. | 10 | 25.9$\times$ | 86.3 | 37.6 | 30.9 |
| $\sqrt{}$ | DYNA. | 10 | 26.1$\times$ | 92.5 | 45.4 | 36.5 |

**Dent helps across attack types.** Table 2 evaluates dent against each attack in the AutoAttack ensemble. Dent improves robustness to each attack type. We report the worst case across these types in the remainder of our experiments.

**Dent helps across datasets and architectures.** We experiment on ImageNet to check scalability. We evaluate the defense of Wong et al. [60], one of few defenses that scales to this dataset, against strong $\ell_\infty$-PGD attacks with 30 iterations, step size of 0.1, and five random starts. Dent improves the adversarial accuracy by 14 points against PGD at $\epsilon_\infty = 4/255$ and natural accuracy by 23 points.

Table **??** in the supplement confirms improvement across more defenses, architectures, and datasets.

## 4.4 Dynamic Defense of Nominal Training

Dent improves the adversarial accuracy of off-the-shelf, nominally-trained models. As dent does not assume adversarial training, it can apply to various models at test time.

For nominal training, we exactly follow the CIFAR reference training in pycls [35, 36] with ResNet-26-4/ResNet-32-10 architectures. Briefly, we train by stochastic gradient descent (SGD) for 200 epochs with batch size 128, learning rate 0.1 and decay 0.0005, momentum 0.9, and a half-period cosine schedule.

We evaluate against $\ell_\infty$ and $\ell_2$ AutoAttack attacks on CIFAR-10. As the nominally-trained models have no static defense, we constrain the adversaries to smaller $\epsilon$ perturbations.

**Dent defends nominally-trained models.** Table 3 inspects how each part of dent affects adversarial accuracy and natural accuracy. When applying dent to nominally-trained models, model adaptation through $\Delta$ is further helped by input adaptation through $\Sigma$. In just a single step, the $\Delta$ update improves adversarial accuracy without affecting natural accuracy. from 8.8% to 15.0% against $\ell_\infty$ attacks with just a single step. With 10 steps, and $\Sigma$ adaptation, dent improves the model's adversarial accuracy to 45.4% against $\ell_\infty$ attacks and 36.5% against $\ell_2$ attacks. In total, dent boosts $\ell_\infty$ and $\ell_2$ adversarial accuracy by almost 40 and 30 points while only sacrificing 3 points of natural accuracy. Dent delivers this boost at test-time, without re-training.

**Input adaptation helps preserve natural accuracy.** Gaussian smoothing significantly improves adversarial accuracy. This agrees with prior work on denoising by optimization [20] or randomized smoothing [7]. Tuned as a fixed hyperparameter, smoothing helps adversarial accuracy but hurts natural accuracy. Optimized end-to-end, our dynamic smoothing reduces the natural accuracy gap. On natural data, the learned $\Sigma$ for the blur decreases to approximate the identity transformation.

Table 4: Adaptive attack by denying updates. We transfer attacks from static models to dent and then evaluate nominal and adversarial training [30] against $\ell_\infty$ and $\ell_2$ AutoAttack. Attacks break the static models (static-static), but fail to transfer to our dynamic defense (static-dent).

Table 5: Adaptive attack by mixing adversarial and natural data. We report the adversarial accuracy on mixed batches, from low to high amounts of adversarial data. Dent improves on adversarial training (43.8%) across mixing proportions within 10 steps.

|  | NOMINAL | | ADVERSARIAL | |
|---|---|---|---|---|
|  | $\epsilon_\infty=\frac{1.5}{255}$ | $\epsilon_2=0.2$ | $\epsilon_\infty=\frac{8}{255}$ | $\epsilon_2=0.5$ |
| STATIC-STATIC | 11.6 | 11.0 | 42.0 | 44.1 |
| STATIC-DENT | 82.5 | 81.6 | 50.0 | 50.2 |

| $\mu, \sigma$ | STEP | 1 | 10% | 25% | 50% | 75% | 90% |
|---|---|---|---|---|---|---|---|
| $\times$ | 1 | - | 43.4 | 43.2 | 44.0 | 44.2 | 43.8 |
| $\times$ | 10 | 62.4 | 51.2 | 49.6 | 48.7 | 48.7 | 47.6 |
| $\checkmark$ | 1 | - | 41.7 | 41.4 | 43.2 | 44.1 | 44.7 |
| $\checkmark$ | 10 | 54.9 | 47.6 | 47.7 | 49.7 | 50.6 | 50.9 |

## 4.5 Adaptive Attacks on Dent Updates

We adaptively attack dent through its use of adaptation by (1) denying updates and (2) mixing batches. To deny updates, we attack the static model offline by optimizing against $\theta$ without $\Delta, \Sigma$ updates, then submit this attack to dent. This attempts to short circuit adaptation by disrupting the first update with a sufficiently strong perturbation. To mix batches, we mix adversarial and natural data in the same batch. This attempts to prevent adaptation by aligning batch statistics with natural data.

**Denying Updates** The aim of this attack is to defeat adaptation on the first move, before dent can update to counter it. We optimize against the static model alone to prevent defensive optimization until adversarial optimization is complete. Under this attack, the input to dent is the final perturbation derived by adversarial attack against the static model.

We examine whether these offline perturbations can disrupt adaptation. Table 4 shows that dent can still defend against this attack. This suggests that updating, and having the last move, remains an advantage for our dynamic defense.

**Mixing Batches** Dent adapts batch-wise, with the underlying assumption that one shared transformation can defend the whole batch. We challenge this assumption by evaluating mixed batches of adversarial and natural data. In Table 5, we vary the ratio of adversarial and natural data in each batch and measure accuracy on the adversarial portion.

At the extreme, we consider an adaptive attack with only one adversarial input per batch. Specifically, we batch one adversarial input with 15 natural inputs randomly chosen from the test set. This adaptive attack aims to reduce adaptation by the dynamic defense, as natural inputs do not need adaptation.

Dent is generally robust to batch mixing, and improves over adversarial training in 10 steps or less.

## 4.6 Ablations & Analysis

**More updates deliver more defense.** The number of steps can balance defense and computation. Table 6 shows that more steps offer stronger defense for both dent and dent+. However, more steps do nevertheless require more computation: ten-step optimization takes $25.9\times$ more operations than the static model (Table 3). As a plus, dent+ is not only more robust, but also more efficient in needing fewer steps. Note that the computational difference between dent and dent+ is negligible, as the adaptation parameters are such a small fraction of the model.

**Model adaptation updates depend on the attack type.** Dent adapts by adjusting normalization statistics and affine transformation parameters. Dent can fix or update the normalization statistics $(\mu, \sigma)$ by using static training statistics ($\times$) or dynamic testing statistics ($\checkmark$); Dent can fix or update the affine parameters $(\gamma, \beta)$ by not taking gradients ($\times$) or applying gradient updates ($\checkmark$). Table 7 compares each combination: affine updates always help, but both updates together hurt $\ell_2$ robustness.

**Batch size** We analyze dent's sensitivity to batch size and focus on small batch sizes. Some real-world tasks, such as autonomous driving, naturally provide a small batch of inputs (from consecutive video

Table 6: Dynamic defenses can trade computation and adaptation. More steps are more robust on CIFAR-10 with $\ell_\infty$ AutoAttack. Dent+ reaches higher adversarial accuracy in fewer steps.

| | STEPS | | | |
|---|---|---|---|---|
| DENT | 0 | 20 | 30 | 40 |
| CARMON ET AL. [5] | 59.5 | 68.3 | 74.7 | 76.1 |
| WONG ET AL. [60] | 43.2 | 48.2 | 52.3 | 55.1 |
| DING ET AL. [12] | 41.4 | 45.4 | 47.6 | 48.7 |
| DENT+ | 0 | 1 | 3 | 6 |
| DING ET AL. [12] | 41.4 | 46.5 | 57.7 | 64.4 |

Table 7: Ablation of model adaptation with and without normalization statistics ($\mu, \sigma$) and affine parameters ($\gamma, \beta$) updates.

| ACCURACY(%) | | NOMINAL | | ADVERSARIAL | |
|---|---|---|---|---|---|
| $\mu, \sigma$ | $\gamma, \beta$ | $\epsilon_\infty = \frac{1.5}{255}$ | $\epsilon_2 = 0.2$ | $\epsilon_\infty = \frac{8}{255}$ | $\epsilon_2 = 0.5$ |
| × | × | 8.8 | 9.2 | 43.8 | 47.3 |
| √ | × | 11.7 | 11.2 | 41.8 | 44.1 |
| × | √ | 16.8 | 16.2 | 49.9 | 57.3 |
| √ | √ | 21.2 | 15.2 | 50.4 | 53.0 |

Table 8: Sensitivity analysis of batch size and adversarial accuracy with dent. With static batch statistics (×), small batch sizes are better. With dynamic batch statistics (√), small batch sizes are worse.

| $\mu, \sigma$ | TYPE | 1 | 2 | 4 | 8 | 16 | 32 | 64 |
|---|---|---|---|---|---|---|---|---|
| × | NAT. | 85.9 | 86.0 | 85.9 | 85.9 | 86.1 | 86.1 | 86.2 |
| × | ADV. | 70.4 | 69.5 | 67.8 | 65.3 | 61.9 | 58.6 | 55.1 |
| √ | NAT. | 11.1 | 68.1 | 76.3 | 80.9 | 83.4 | 84.9 | 85.8 |
| √ | ADV | 5.8 | 35.9 | 48.3 | 53.0 | 55.3 | 54.4 | 52.9 |

frames or various cameras, for example), and so we confirm that dent can maintain robustness on such small batches. Table 8 varies batch sizes to check dent's natural and adversarial accuracy.

# 5 Discussion

In advocating for dynamic defenses, we hope that test-time updates can help level the field for attacks and defenses. Our proposed defensive entropy method takes a first step by countering adversarial optimization with defensive optimization over the model and input. While more test-time computation is needed for the back-and-forth iteration of attacks and defenses, the cost of defense scales with the cost of attack, and some use cases may prefer slow and strong to fast and wrong.

**Limitations** Dent depends on batches to adapt, especially for fully test-time defense without adversarial training. It also relies on a particular choice of model and input parameters. A different objective could possibly lessen its dependence on batch size and reliance on constrained updates. More generally, dynamic defenses may present difficulties for certification or deployment, as they could drift. Along with how to update, improved defenses could investigate when to reset, or how to batch inputs for joint optimization.

**Benchmarking** Standardized benchmarking, by AutoAttack and RobustBench for example, drives progress by competition and empirical corroboration. Dent brings adversarial accuracy on their benchmark within 90% of natural accuracy for three of the most accurate methods tested [5, 61, 12]. This is encouraging, but more research is needed to fully characterize dynamic defenses like dent. However, RobustBench is designed for static defenses, and disqualifies dent by its rule against test-time optimization. Continued progress could depend on a new benchmark to standardize rules for how attacks and defenses alike may adapt.

By fighting gradients with gradients, dent shows the potential for dynamic defenses to update and counter adversarial attacks. The next steps—by attacks and defenses—will tell.

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
