# OpenReview forum: "Fighting Gradients with Gradients: Dynamic Defenses against Adversarial Attacks"
_NeurIPS.cc/2021/Conference — NeurIPS 2021 Submitted_

### Official Review · Reviewer_jWma · 2021-07-06

**Rating:** 4
**Confidence:** 4

**Summary:**

The paper proposes a novel test-time dynamic defense method, which adapts the model and input during testing by defensive entropy minimization (named dent). Given an input, the proposed method minimizes the entropy of the model prediction to adapt the input and updates parameters in batch normalization layers. Empirical results show that the proposed method improves the robustness of adversarially/nominally-trained models against various attacks including AutoAttack.

**Limitations And Societal Impact:**

The authors have addressed limitations of the proposed method in Section 5.

The authors have not described negative societal impacts since they propose a defense method.


**Main Review:**

Pros:
- The idea is novel and very interesting.
- The paper is easy to follow.
- Experimental results show strong robustness against existing attacks including AutoAttack.
- The paper considers adaptive attacks.

Cons:
- There is a big concern about vulnerability against a kind of data poisoning attacks at test time.

-------------
Overall, this work is interesting and impressive. The idea is novel and the strong robustness against AutoAttack provides useful information for researchers in this field.

However, there is a big concern about vulnerability against a class of adaptive attacks. The proposed method sequentially updates model parameters, and it is concerning that this type of defense methods could be vulnerable against a kind of data poisoning attacks at test time. Before attacks, between attack steps, and a part of each batch, attackers can feed malicious inputs (e.g., random noises) to make model parameters meaningless. The attackers can also craft inputs to broke model parameters.

Although this type of attacks is not used in existing attacks, it is because the parameters of the target model is typically fixed, and these attacks are meaningless against the fixed target models. In white box settings, while the purpose of attackers is to find an adversarial example with small perturbation, attackers can feed any inputs to the model for their purpose. Is the proposed defense robust against these attacks? It is important to present discussions and experiments for this type of attacks to show the robustness of the proposed method in white-box settings.


I would be happy to raise my score in case these points are properly addressed.

Other comments:

- The word “step” is sometimes confusing. There are two kinds of steps in the paper: attack steps and dent steps. For example, Figure 1 shows three attack steps for static/dynamic defenses, while “Step” in Table 3 is dent steps (i.e., steps to update $\Delta$ and $\Sigma$). However, it seems no sufficient explanation about dent steps in the paper.

- The drawback of dent+ is not clear. In line 148, the authors wrote “Dent adapts on batches rather than samples. Batch-wise adaptation stabilizes optimization”, but Table 1 shows that sample-wise Dent+ achieves better performance. I feel the statement lacks consistency. Readers may think that dent is not necessary as dent+ is better than dent.

- The difference in the optimizer for Dent and Dent+ is a bit weird. Why do the authors adopt AdaMod with large learning rate for Dent+, and how do you decide the optimizer?


**Time Spent Reviewing:**

6 hours

---

> ### Author Response · Authors · 2021-08-11
> **Parameter Updating, Poisoning, Dent vs. Dent+, the Meaning of "Step", and Optimizer Ablation**
>
> We thank the reviewer for considering dent "interesting and impressive" and for their thoughts on the potential for a new type of data poisoning attack.
>
> **"The proposed method sequentially updates model parameters"**
>
> Dent resets between batches (l. 149). Dent does iteratively update on each batch, but the inputs in a given batch can only alter the predictions for that batch. That is, one batch cannot affect another batch.
>
> **"this type of defense methods could be vulnerable against a kind of data poisoning attacks at test time"**
>
> We thank the reviewer for their intriguing idea about poisoning dynamic defenses. Could the reviewer please elaborate on how such a poisoning attack could be more successful than a standard direct attack? We evaluate in the white-box setting (Section 4.2) with norm bounds, so the poisoner would have the same constraint as the attacker. Because dent preserves independence of predictions across batches (as highlighted above), a poisoner would have to poison each and every batch. The suggested use of random noise would surely violate the norm constraint, no?
>
> Please let us know if poisoning is still a concern after the clarification that dent resets between batches.
>
> **dent vs. dent+**
>
> The reviewer is correct in that dent+ is an improvement on dent. Allow us to clarify batch-wise and sample-wise adaptation. Dent and Dent+ both require batches of data and update the affine parameters and statistics of normalization layers (lines 127-129). They differ in the dimension of the scale and shift parameters (lines 135-137). For batch dimension N and channel dimension K, dent updates 1xK parameters while dent+ updates NxK parameters, which the text refers to as "batch-wise" and "sample-wise" parameters. The mean and variance statistics for dent and dent+ are 1xK, and therefore depend on the batch. In this way dent+ updates batch-wise and sample-wise.
>
> We will clarify these parameters by explicitly stating the dimensions, as we do in this response.
>
> **“step” is sometimes confusing**
>
> Thank you for identifying the "overloading" of "step" between attacks and defenses. We will edit to disambiguate "steps" of attack and defense vs. "updates" by dent (that is, iterations of optimization.
>
> **Optimizer for Dent and Dent+**
>
> We tuned hyperparameters on PGD attacks (l. 174-175). Empirically, dent requires more steps for improvement so we reduced the learning rate, while dent+ requires fewer steps so we increased the learning rate. To double-check the effect of AdaMod as requested, we re-run dent+ with adam:
>
> adam with lr 1e-3 (default lr for adam): wong2020fast static 43.21 vs dynamic 55.40, ding2020mma static 41.44 vs dynamic 55.28
> adam with lr 6e-3 (same lr as adamod): wong2020fast static 43.21 vs dynamic 71.9, ding2020mma static 41.44 vs dynamic 60.03
>
> Dent+ with these settings is still better than dent (see Table 1). We will include this ablation in the supplement because the learning rate and AdaMod can help.

---

### Official Review · Reviewer_tNPw · 2021-07-12

**Rating:** 5
**Confidence:** 3

**Summary:**

This paper proposed a new defense, namely, defensive entropy minimization (dent), to improve adversarially robustness. The proposed method alters testing but not the training phase, which is different from most existing defenses that are static and do not adapt during testing. The improvements of model robustness evaluated by different strong attacks and datasets have confirmed its effectiveness.

**Limitations And Societal Impact:**

As aforementioned, could the authors further lessen the method's dependence on the batch size of test data? (or give more discussion?)

**Main Review:**

Pros:
1. This paper is clearly organized and easy to follow.
2. The proposed method (dent) is interesting and reasonable.
3. Extensive experimental results on multiple datasets are provided and the effectiveness of the proposed method is confirmed.

Cons:
1. The performance of the proposed method (dent) is not stable as the static defense (e.g., adversarial training). When the batch size of the test data becomes smaller, the defense effect will be weakened. Since the adaptation is based on the test data, dent may not be able to handle the test scenarios (e.g., using small test batch size or hard dataset like ImageNet) which are not friendly to the estimation for the sample-wise parameters. Could the authors give more discussion or analysis and make some improvements to the method?
2. The relationship between this method and domain adaptation is not very clear. Could the authors give more explanation about it and how dent improves robust accuracy while maintaining natural accuracy?
3. Lack of further analysis or discussion on the performance using different datasets. More experimental results or discussion about dent's performance on the large dataset (i.e., ImageNet) are encouraged.

Minor Concerns/Questions:
1. It seems that the correct reference is missing in line 224. (The "Table ??" in line 224)

**Time Spent Reviewing:**

6

---

> ### Author Response · Authors · 2021-08-11
> **More Datasets, Batch Dependence, and the Relationship to Domain Adaptation**
>
> **More Datasets: "performance using different datasets" and "dent's performance on the large dataset (i.e., ImageNet)"**
>
> We report AutoAttack results with CIFAR-100 in Table 9 of the supplement (referenced on l. 224 without the table number due to a typo). We report an ImageNet result against PGD on lines 220-223 of the main text: dent improves both adversarial and natural accuracy for adversarial training by Wong et al. [60]. We choose CIFAR-10 for most of our experiments as it is the focus of RobustBench and many exising works.
>
> Please let us know if CIFAR-100 and ImageNet are the different datasets indicated by the reviewer.
>
> **"could the authors further lessen the method's dependence on the batch size of test data? (or give more discussion?)"**
>
> We acknowledge the dependence on batch size throughout the text (Fig. 2, l. 148-149, l. 199-202) and signal it at the beginning of the limitations (l. 290).
>
> We analyze sensitivity to batch size in Section 4.6 (Table 8 and l. 280-283) for dent. This sensitivity is inherent to batch normalization statistics. For Table 8, the reference result is 43.8% for the static defense of Adversarial Training by Madry et al [31]. Dent improves robustness for batch sizes >= 4 even with batch-wise statistics updates (bottom two rows). We agree with the reviewer that no batch dependence at all would be better—we motivate research in this direction by our results with dent and improved results with dent+.
>
> Dent updates purely batch-wise while dent+ introduces sample-wise parameters. Dent+ adapts the scale and shift parameters for each input instead of sharing the parameters across the batch (lines 134-137 describe the parameters, lines 213-216 summarize the result for dent+, and Table 6 compares batch-wise dent and sample-wise dent+). Dent+ shows results can be improved and batch dependence lessened, but there is more work to be done.
>
> **"The relationship between this method and domain adaptation is not very clear"**
>
> Inspired by test-time domain adaptation to dataset shift [43, 53, 58], dent improves the robustness of static defenses by adapting differently to attacked and clean data. This can be seen in dent improving robust accuracy while not reducing natural accuracy: it adapts differently to the two types of data. If not, the results would not differ across the types.

---

### Official Review · Reviewer_B1uw · 2021-07-14

**Rating:** 6
**Confidence:** 3

**Summary:**

This paper studies the problem of dynamic defenses agaisnt the potential adversarial attacks during the test time. The work introduces a new method derived from "defensive entroy minimization" approach that "fights back" and counter-optimize against attacks. In general, the proposed method updates the parameters of DNN slightly for every round of inferences, so as to balance the accuracy  and fuzziness of models. The basic idea was clear while experiment results demonstrated the potentials of proposed methods. During the experiments, authors aimed at establishing three major claims (1) Dynamic Defense of Adversarial Training, (2) Dynamic Defense of Nominal Training, and (3) Adaptive Attacks on Dent Updates, with sub-claims in details. Experiment results support these claims in part. Ablation studies also demonstrate the indispensibility of and contributed made made by every component.

**Ethics Review Area:**

["I don’t know"]

**Limitations And Societal Impact:**

My major concern on this paper is that how dent ensures the parameter update \Delta would not hurt the performance of neural networks. As was learned in some previous (e.g., lottery ticket theory), the modification to the weights might not affect the way that DNN behave. Or we can say that updating weights might not be the most efficient way to achieve the goal. Just be curious, have you explored the potentials of feature maps perturbation, where the behaviors of the DNN could be slightly changed with accuracy of prediction preserved?

**Main Review:**

Typo: 224 "Table ?? in the supplement..."
The work was comprehensive and intuitive with details in experiments and comparisons.
Supplementary materials provided necessary information to follow this work.

**Time Spent Reviewing:**

3 hrs

---

> ### Author Response · Authors · 2021-08-11
> **Ensuring Dent Helps, Updating Parameters vs. Feature Perturbations**
>
> **"how dent ensures the parameter update \Delta would not hurt the performance"**
>
> In principle, dynamic defenses such as dent could hurt accuracy as their test-time updates are not supervised. In practice, we check that dent does not hurt by verifying the accuracy of dent+ on natural/clean data: standard test accuracies with and without dent+ are within 1% absolute. Please see the supplement for the summary of these results in README.md. Dent's choice of entropy as its defense objective helps avoid this potential issue: predictions tend to have high confidence/low entropy when the data is not attacked or shifted, and so there is no need to update, and dent does not hurt accuracy. Please see the prior work on Tent [58] for more analysis of entropy w.r.t. non-adversarial shifts like corruptions (especially Figures 2 and 5).
>
> **"have you explored the potentials of feature maps perturbation"**
>
> Our choice of adaptation parameters for the model (lines 127-133) corresponds to channel-wise affine perturbation of the feature maps. More specifically, updating the batch norm scale and shift parameters along with the mean and variance statistics yields different scalings and shiftings of the feature channels that depends on the test input. We note that these are a "small portion of the full model parameters" in the text, and can edit this paragraph to make the connection with perturbations.
>
> **Typo: 224 "Table ?? in the supplement..."**
>
> Please note that Line 224 should refer to Table 9 in the supplement. This table reports results for Adversarial Training [31] and TRADES [67] with CIFAR-10 and CIFAR-100 and ResNets of different depths/widths. (We apologize for the typesetting error which we have now corrected.)

---

### Official Review · Reviewer_XQWL · 2021-07-15

**Rating:** 4
**Confidence:** 5

**Summary:**

This paper proposes to defend deep networks by dynamically modifying the model and the input images. The model and the input are modified by optimizing the Shannon entropy with respect to the model and the input.

**Limitations And Societal Impact:**

This work does not have obvious negative societal impact. I have listed the limitations in the main review part.

**Main Review:**

Major Issues:
1) First of all, modifying the input [1,2] and dynamically modifying the model [3] for defensive purpose has been explored in previous works. The authors should cite these works and be more careful and precise with the claim of :
> Line 53 We demonstrate the **first** defense to optimize the model and input during testing for improved robustness.

2) The key design of this work is the proposal of minimizing the Shannon entropy, which confuses me a lot. How come can optimizing the current prediction help find the correct prediction? For an adversarial example that is wrongly classified, wouldn't optimizing the wrong prediction worsen the prediction. To be more clear, from my experience, for an example with label A, if adversarial attacks can modify the example to make the model predict B, and the dent method optimizes the entropy for the adversarial example, it is likely to be in favor of B rather than A. After briefly reading the provided code of cifar10a.py, I think I find a possible reason. In the line 28 of cifar10a.py, the adversarial examples are generated on a dynamic model rather than a static one. The dynamic model may hamper the attack to find a proper adversarial example. Thus, I have two questions:
* What is the accuracy of the attack in the line 28 and does the attack really succeed? This means what is the accuracy if adversarial examples are generated with dynamic model but predict by a static model? If the adversarial example generated on the dynamic model can not fool the static model, then this is an evidence that the attack in the code is not properly implemented.
* What will happen if we generate adversarial examples on a static model then predict these adversarial examples on dynamic one? I suspect that, in this setting, the dent method will be broken. BTW, to show the effectiveness, [1] also tested on both static and dynamic/randomized model. The author should follow the same routine.

Without clarifying these facts, it is very likely that the huge improvement is simply an illusion of incorrect attacks implementation.
Also notice that, the author claims:
> Line154 : Our defense does not rely on shattered gradients

This claim is questionable. It seems to me that the dynamic process is not differentiable and may hamper a proper gradient computation. The author should adopt the BPDA in [4] to show it does not suffer from the obfuscated gradient. Also, the author claims that their method is deterministic and thus does not suffer from stochastic gradient problem (Line155). This is somewhat confusing. If the method is **deterministic**, then this method should not be claimed as **dynamic**. Please explain.

3) The evaluations in this work is questionable on several aspects. For robust models, the author omit two of the most important works: TRADES[5] and Adversarial Training[6]. Although the models tested in this paper are occasionally built up on these two methods, the author still should provide evaluations on these two important baselines for better comparison. Also, on Table 2, why only provide the detailed results on only two methods? Clearly the authors have the information for other models since they show it on Table1. The presented two methods Wong and Ding are either not the most representative nor have the best performance.

4) As claimed by the author, the methods require batch predictions. What causes this deficit? To me, the modification of the model should not rely on batch information so much since it is stable after training. On Table 8, the author says:
> With static batch statistics (×), small batch sizes are better. With dynamic √batch statistics ( ), small batch sizes are worse.

What causes this phenomena?

Minor Issues:
1) For Line 29~32, I understand that adversarial robustness is related to domain adaptation. But how does this fact relate to the method in this work?
2) For Line 36, what does "adaptive attacks that are aware of its updates" refer to
3) For Line 135~137, please be more clear? Have dent solve the batch-wise deficiency?
4) For Line 158, what does "stale gradient" mean?
5) I recommend the author to write an algorithm to show their method more clear.
6) For Line 224, Table what?


The second point of the major issues is my major concern about this work. I will change my score if the author can answer my questions and provide solid proofs for their effectiveness.


[1] Mitigating adversarial effects through randomization.    ICLR 2018

[2] Randomization matters How to defend against strong adversarial attacks    ICML 2020

[3] Adversarial Robustness via Runtime Masking and Cleansing    ICML 2020

[4] Obfuscated Gradients Give a False Sense of Security: Circumventing Defenses to Adversarial Examples    ICML 2018

[5] Theoretically Principled Trade-off between Robustness and Accuracy    ICML 2019

[6] Towards Deep Learning Models Resistant to Adversarial Attacks

**Time Spent Reviewing:**

8

---

> ### Author Response · Authors · 2021-08-11
> **Novelty, Validity of AutoAttack and Adaptive Attacks, Dynamic vs. Deterministic**
>
> **Novelty of Dent and Reviewer's References [1, 2, 3]**
>
> Thank you for providing these references.
>
> Our claim (l. 41-24) is that
>
> > We propose the first *fully test-time dynamic defense*: dent adapts both the model and input *during testing without needing to alter training*
>
> which contrasts dent with [1, 2, 3]. Adapting is not the same as modifying: [1,2] modify the input by randomization but do not adapt. Their randomization is independent of the input and so they do not optimize to condition on it. [3] optimizes on test data, but alters training in their "prepare" step, and still requires all of the training data in their "predict" step at test time. Dent only needs the model and test data, which is what makes it "fully test-time" and is what we mean by "during testing." This aligns with the existing terminology of test-time adaptation [53, 58].
>
> Thank you for raising this point—we will edit l. 53 in the related work to mirror the "fully test-time" claim in the introduction. We will discuss and cite RMC [3] in the related work as it does update during testing but unlike dent also alters training.
>
> **Validity of Attacks: "the huge improvement is simply an illusion of incorrect attacks implementation"**
>
> We are in complete agreement on the importance of rigorous evaluation. Thank you for inspecting our code and considering the transfer of attacks between static and dynamic models!
>
> For our evaluation with standardized attacks, we stress that we evaluate with the official benchmark version of AutoAttack, so our attacks cannot be incorrect w.r.t. the reference implementation. (Please refer to this point in our shared response).
>
> For our evaluation with adaptive attacks across static and dynamic models, the static-dynamic direction is included in the paper and we report the dynamic-static direction as requested by the reviewer:
>
> **"if we generate adversarial examples on a static model then predict these adversarial examples on dynamic one?"** This is our "denying updates" adaptive attack (l. 253-259) with results in Table 4. In the static-dent case, when the attack is made against a static model and then predicted with the dynamic model, dent still improves the robustness of the static model alone (50.0% vs. 42.0%). The static defense is [12].
> **"if adversarial examples are generated with dynamic model but predict by a static model?"** We experimented with this direction at the reviewer's request. When the APGD-CE attack is made against dent+ then transferred to the static model, the robustness improves to 61.4% vs. 50.1%. As above, the static defense is [12].
>
> Thank you for the suggestion—we will expand Table 4 to include dynamic-to-static along with static-to-dynamic.
>
> **"If the method is deterministic, then this method should not be claimed as dynamic"**
>
> We define "dynamic defense" as adapting during testing (l. 22) and "dynamic model" as conditioning on the input (l. 85). Whether the method is deterministic vs. stochastic is not key to our definition. A randomized defense that adds noise during testing is stochastic but not dynamic because the noise is independent of the input (l. 20-21). Our dent defense conditions on the input by updating model parameters and statistics, so it is dynamic, but not stochastic because its updates rely only on the input and static model parameters. There is no randomness in its forward or backward passes. Given the exact same input, dent will always make the exact same update.

---

> ### Author Response · Authors · 2021-08-11
> **How Entropy Helps, Evidence against Gradient Obfuscation, Choice of Models, Stale Gradients, and Minor Comments**
>
> **How Entropy Helps: "The key design of this work is the proposal of minimizing the Shannon entropy, which confuses me a lot"**
>
> Dent updates by taking gradients of entropy and estimating batch statistics, and together these updates help defense. Its updates make inputs in the same batch dependent, and we hypothesize that attacks are unstable w.r.t. this batch dependence. Additionally, we note that entropy is a function of all logits, and so the gradient may be sharper or flatter for the true or attacked class. As these are conjectures, we did not include them in the text, but we could include these possibilities in the discussion if they are informative.
>
> The reviewer is right, in that if the attacker already had an attack that made dent misclassify an input with label A as label B, then further dent updates would only make the misclassification worse.
>
> In practice, dent's updates make it more difficult to find such an attack, and transferring from a static model without dent's updates is not enough to nullify its improved robustness. The results of Table 4 on transferring attacks from a static model to a dynamic model show that it is not trivial to circumvent these updates. In this case, dent remains more robust than the static model alone.
>
> **"to show it does not suffer from obfuscated gradient"**
>
> Thank you for highlighting gradient obfuscation. We agree on its importance, and so we discuss it on lines 153-157. Please allow us to clarify the reviewer's concerns on differentiability and stochasticity, and then list evidence against obfuscation from our results.
>
> - "It seems to me that the dynamic process is not differentiable" Dent is differentiable. Dent makes its own gradient updates during testing, so our defense in fact requires differentiability to adapt model and input parameters.
> - "should adopt the BPDA" Dent updates are differentiable, so we do not apply backward pass differentiable approximation (BPDA) as the exact gradient is available.
> - "suffer from stochastic gradient" Dent is deterministic: given the exact same input, dent will always make the exact same update.
>
> The results do not show signs of obfuscation:
>
> 1. White-box attacks do better than black-box attacks (Table 2).
> 2. Attacks do better with larger norm bounds (Figure 4, supplement).
> 3. Attacks do better with more iterations (Tables 13 & 14, supplement).
> 4. Attacks without norm bounds reach 100% success (Figure 4, supplement).
>
> **Choice of Robust Models: "omit two of the most important works: TRADES[5] and Adversarial Training[6]"**
>
> Results with Adversarial Training (by Madry et al.) [31] and TRADES [67] are in Table 9 of the supplement. The same pattern holds, with improvement from dent across CIFAR-10 and 100 and ResNet-26-4 and ResNet-32-10. We apologize for the table reference typo on l. 224 of the text. We thank the reviewer for underlining the importance of these baselines, and we will explicitly mention them in the revision when referring to the supplement.
>
> **"on Table 2, why only provide the detailed results on only two methods?"**
> It is simply for space. We chose [12, 60] because they are the most efficient, and fastest for others to verify with our code. We will add the full attack-wise results for each row of Table 1 to the supplement and thank the reviewer for the suggestion. Table 2 is included to offer attack-wise detail beyond the summary in Table 1, but we include more rows in Table 1 to more thoroughly compare static, dent, and dent+ defenses.
>
>
> **what does "stale gradient" mean?**
>
> The gradient for the attack at time t is "stale" because it used the model parameters at time t-1. Since dent is dynamic, the model parameters update on each input, so the attack is always one step behind and the defense has the last move. Please refer to the discussion on lines 150-152 alongside line 158.
>
> **"Have dent solve the batch-wise deficiency?"**
>
> Dent and Dent+ both require batches of data and update the affine parameters and statistics of normalization layers (lines 127-129). They differ in the dimension of the scale and shift parameters (lines 135-137). For batch dimension N and channel dimension K, dent updates 1xK parameters while dent+ updates NxK parameters, which the text refers to as "batch-wise" and "sample-wise" parameters. The mean and variance statistics for dent and dent+ are 1xK, and so depend on the batch.
>
> **"adaptive attacks that are aware of its updates"**
>
> Section 4.5 covers adaptive attacks that specifically interfere with dent's adaptation, which require knowing that dent is updating and how it updates. Note that all attacks in our evaluation, including AutoAttack, are aware of dent's parameter and statistics updates in the sense that they have access to the exact predictions and gradients of the adapted model (which are a function of the updated parameters and statistics). This reflects the standard white-box setting (lines 195-198).

---

### Official Review · Reviewer_K7sr · 2021-07-28

**Rating:** 5
**Confidence:** 4

**Summary:**

This paper provided a new perspective on adversarial defense, i.e., optimizing adversarial defenses at inference.

The authors proposed dynamic defenses that adapt the model and input during testing to defend against adversarial attacks.

**Limitations And Societal Impact:**

see above.

**Main Review:**

Pros:

1 The new perspective on defense at inference time indeed deserves some explorations. This perspective is interesting.

2 comprehensive experiments on CIFAR-10/100 and ImageNet are reported; the experimental results look good.

Cons:

1 Auto-attack is meant for evaluating a single & static model, which may not be applicable to evaluating dynamic defense. Therefore, authors may consider the adaptive attacks for the robustness evaluations.

2 Inference-time defense may hurdle the speed of the model prediction.

Some other comments:

The authors implicitly assume adversarial data comes from a different distribution from that of natural data. Could authors elaborate more about it?

* I may adjust my scores based on the authors' responses.


#### Post rebuttal ####
The authors did not resolve my concerns. Please find my reasons as follows.

Auto-attack is meant for evaluating a single & static model, which may not be applicable to evaluating dynamic defense. Therefore, using AA attacks for evaluating the dynamic defense is a factual error.
Although there are some discussions of adaptive attacks on "dent", those adaptive attacks strategy still cannot convince me enough.



**Time Spent Reviewing:**

4 hours

---

> ### Author Response · Authors · 2021-08-11
> **AutoAttack + Adaptive Attacks, Inference Time, and Adversarial vs. Natural Data as Domains**
>
> **Use of AutoAttack**
>
> We evaluate with AutoAttack as it is the current standard benchmark. We discuss benchmarking in Section 5, and point out the lack of (and opportunity for) a standardized benchmark for dynamic defenses. We hope to motivate research on such a new benchmark in having shown a large margin between dynamic and static defenses against AutoAttack.
>
> **Adaptive Attacks**
>
> Our adaptive attacks are introduced on lines 191-194 and covered in Section 4.5. To summarize, we propose and experiment with two adaptive attacks against dent's dynamic adaptation: (1) denying updates, so the attacker can first optimize against the static model without our defense (2) mixing data, so the attacker can combine adversarial and natural data in the same batch. Both interfere with the gradient and statistics updates by dent, but it still proves more robust than a static model. Please see Tables 4 and 5 in the paper.
>
> **Inference Time**
>
> Our dynamic defense is more robust than a static defense but it does take more time. We claim that dynamic defenses can be more robust, not more efficient during inference, and we give the user control over the amount of computation to balance robustness and efficiency.
>
> We discuss computation and defense on lines 269-274 and measure the trade-off in Table 6. One update step by dent+ gives a 5 point improvement in robust accuracy in ~3x the computation of the forward pass by the static model. At six steps the robust accuracy improves by >20 points absolute.
>
> We explore reducing inference time for dynamic defense with our dent+, which needs only 0.1x the steps of dent. Further improving efficiency of dynamic defenses is an important direction for future work.
>
> **"assume adversarial data comes from a different distribution from that of natural data"**
>
> We are guided by prior research such as Madry et al. [31] and Rice et al. [38] that find more capacity can help adversarial robustness and furthermore that it is possible to overfit to adversarial data. If natural data and adversarial data were from the same distribution, then more capacity should not help, and selective overfitting to one or the other should not be possible. As more direct evidence, consider our results: dent does not change natural accuracy but does change robust accuracy. To have a different effect, dent must update differently on each type of data.

---

### Author Response · Authors · 2021-08-11
**Shared Response to All Reviewers**

We thank all of the reviewers for their valuable feedback, questions, and acknowledgements of our contributions. We address common points in this shared response, and then respond to each reviewer individually.

**Dynamic Defenses and Adaptation (XQWL, tNPw)**

Dent is a new defense that updates during testing to improve the robustness of state-of-the-art defenses against the standard and strong AutoAttack benchmark by a large margin (30% relative). It is *dynamic* in that it *adapts* to the input during testing, unlike *static* defenses that only optimize during training. This sets it apart from adverarial training defenses, which are constant during testing, and randomization defenses, which modify the input but do not adapt to it. Furthermore, dent is unique among defenses in updating the model fully at test time—it neither alters training nor requires the training data for its updates during testing.

This definition of dynamic directly connects to adaptation: a defense is dynamic if it adapts by conditioning on the test input. It is updating that makes a method dynamic, even if the update is deterministic, as is the case for dent's gradient updates according to entropy and statistics updates according to batch-wise means and variances. By updating during testing, dent can adapt differently to attacked data and natural data, which relates defense to dataset shift in domain adaptation.

**Evaluation by Standard Attacks (XQWL, K7sr)**

We stress that our AutoAttack evaluation relies on the reference implementation that is part of RobustBench, a standardized benchmark. We do not modify the attacks in any way. Our defense obeys the standard defense interface: it takes batches of test inputs and makes batches of predictions without any other data/privileged information/etc. We underline that dent resets between every input batch (l. 149), so it truly is only a function of the current batch. Yes, our dynamic defense updates its parameters while making predictions, so the attacks are against a dynamic model as XQWL points out. That is the very purpose of a dynamic defense.

The reason for this evaluation is to show dent's effect in standardized setting for comarison with existing work. We agree with the reviewers that AutoAttack is designed for static defenses, as we discuss (Section 5). That is why we also evaluate by adaptive attacks against our dynamic updates (Section 4.5).

**Evaluation by Adaptive Attacks (K7sr, XQWL, jWma)**

As a dynamic defense, dent depends on the test data, so we propose and evaluate two adaptive attacks to try and exploit this dependence, and we also ablate its sensitivity to changes in its test-time updates (number of updates, types of updates, and batch size). Section 4.5 covers our adaptive attacks and Section 4.6 covers analysis and ablations specific to dent's adaptation.

More and new attacks are always possible. Our adaptive attacks (1) deny updates and (2) mix batches because of dent's specific use of optimization and batch statistics. It is our intention to call attention to dynamic defenses so that more effort is attracted to adaptive attacks and benchmarking (Section 5, l. 296-302). Our results do so by showing dent's continued robustness to adaptive attacks that target its particular updates and dent's strong effect on robustness to standardized attacks.

**Supplementary Material (XQWL, B1uw, tNPw)**

Please note that Line 224 should refer to Table 9 in the supplement. This table reports results for Adversarial Training [31] and TRADES [67] with CIFAR-10 and CIFAR-100 and ResNets of different depths/widths. (We apologize for the typesetting error which we have now corrected.)

The purpose of our work is to highlight dynamic defenses and promote discussion, and so we would be happy to engage with reviewers during the discussion phase if there are further points to address. Thank you.

---

### Decision · Program_Chairs · 2021-09-27

**Decision:**

Reject

**Comment:**

The paper proposed to dynamically modify the model and the input images during the test time (named "defensive entropy minimization"). While the idea is interesting, the true novelty needs to be carefully clarified given many missing related papers, because the test-time defense is in fact not new but one of the two main-stream approaches to adversarial robustness (the other is adversarial training, i.e., training-time defense by simulating the presumed attack without seeing the real attack). Besides the first 3 papers given by Reviewer XQWL, there is another paper closely related to test-time defense: Countering Adversarial Images using Input Transformations, ICLR 2018 (https://arxiv.org/abs/1711.00117). The paper cannot be accepted this time, and hope the authors can find the information helpful for their submission next time!